# Molecularly Imprinted Electrochemical Sensor Based on Poly (O-Phenylenediamine) for Sensitive Detection of Oxycodone in Water

Pranaya Charkravarthula and Amos Mugweru *

Department of Chemistry and Biochemistry, Rowan University, Glassboro, NJ 08028, USA
* Correspondence: mugweru@rowan.edu; Tel.: +1-856-256-5454; Fax: +1-856-256-4478

**Abstract:** This work was aimed at the development of a sensitive electrochemical detection method for oxycodone in water. Molecularly imprinted electrodes were formed by electro-polymerization process using o-phenylenediamine as a monomer. The electro-polymerization was performed on glassy carbon electrodes in the presence of oxycodone before the extraction of entrapped oxycodone molecules. Various electrochemical techniques were employed to monitor the polymerization and response of the fabricated electrodes toward oxycodone. These techniques included cyclic voltammetry (CV), square wave voltammetry (SWV), differential pulse voltammetry (DPV) and electrochemical impedance spectroscopy (EIS). The oxycodone concentration was determined using SWV by measuring the change in the oxidation peak current of $[Fe(CN)_6]^{3-/4-}$ in a 0.1 mM acetate buffer solution. At the optimal electro-polymerization conditions, a calibration curve of the current versus the concentration of oxycodone indicated a linear response at a region from 0.4 nM to 5.0 nM with a detection limit of $1.8 \pm 0.239$ nM. The MIP-modified electrode's binding isotherm was fitted using a Langmuir model and showed an association constant, $K_A$, of $1.12 \times 10^6$, indicating a high affinity of oxycodone molecules to binding sites. This sensor has the potential to act as an alternative method suitable for the on-site analysis of oxycodone.

**Keywords:** oxycodone; electro-polymerization; square wave voltammetry; cyclic voltammetry; molecular imprinting; o-phenylenediamine

## 1. Introduction

There is an urgent health concern related to opioid overdoses in the world, with the fatality rate reaching alarming proportions [1]. Oxycodone, one of the opioids, is a semi-synthetic narcotic analgesic. It is also marketed in combination with products such as aspirin or acetaminophen. Oxycodone is commonly used to relieve pain, especially for those patients undergoing surgical procedures [2]. Oxycodone overuse can result in other conditions, including addiction. In cases of oxycodone overdoses, death due to respiratory depression can occur. It is necessary to monitor the concentrations of oxycodone in the blood and urine of patients. Therefore, sensing devices/tools that enable rapid measurements of opioids are needed.

In vitro oxycodone metabolites include oxymorphone and noroxycodone, which have no pharmacologic effects. Figure 1 shows the structures of common oxycodone metabolites. The primary method to measure oxycodone, as well as other opioids, is HPLC–MS [2–4]. The technique is sensitive and provides extreme specificity and low limits of quantification of oxycodone. However, HPLC–MS is quite expensive, and professional technicians are needed for the complicated operations to fulfill the wide detection demands. The technique also takes a few hours, limiting immediate clinical decision making. Therefore, techniques that offer a fast response time, high sensitivity and simple usability would be preferred. Searching for cheap, stable and sensitive devices for opioid detection is very important.

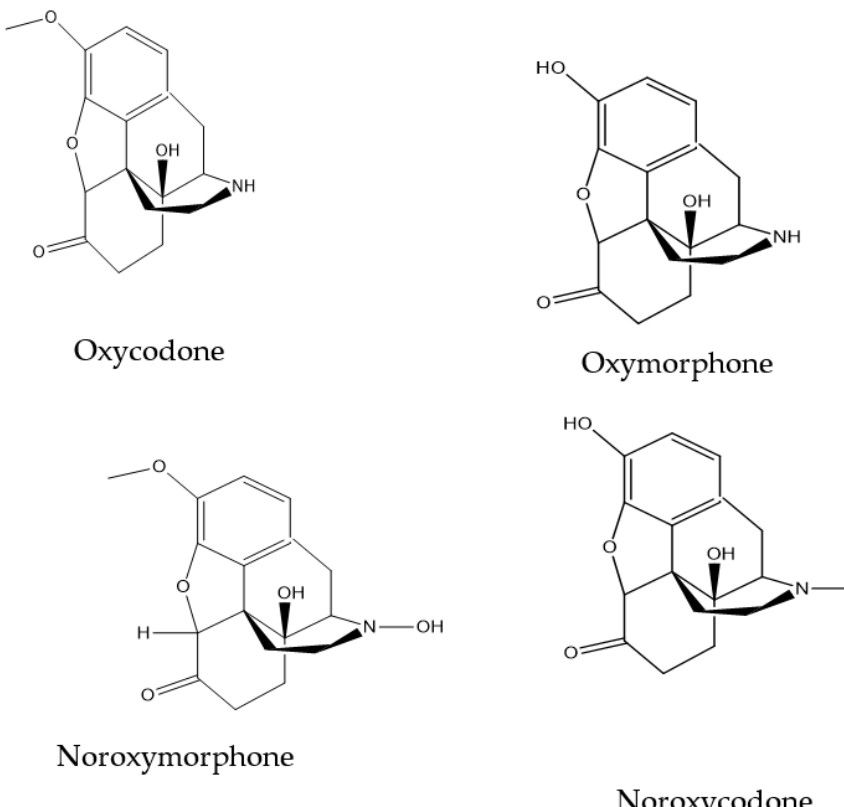

**Figure 1.** Oxycodone molecular structure and some of its oxidation metabolites.

Many simpler opioid-sensing platforms have been developed, including those using nanomaterials, such as gold nanoparticles, silver nanoparticles [5,6], thiourea-doped graphene [7] and carbon nanotubes [8]. Biological-based receptors for use in chemo-/biosensors are very important. Those that use synthetic counterparts as a recognition element have especially gained traction for chemical analysis. Particularly, the molecular imprinting technique combined with electrochemical detection has become a powerful tool to selectively bind and determine chemical species [9–13]. These molecular imprints serve as tailorable, synthetic molecular recognition platforms. These imprints are polymeric receptors capable of binding specific molecules, akin to the "lock-and-key" mechanism. Their ease of preparation, as well as the potential for them being suitable for on-site analysis, make them attractive methods for the creation of highly efficient synthetic molecular receptors. Molecular imprints are therefore superior compared to antibodies or enzymes because of their chemical stability under variable conditions, including high temperatures [14]. However, one major detriment to the development of MIP-based materials for sensors is the occurrence of nonspecific binding. The extent of nonspecific binding is determined by the kind of polymer used in imprinting.

Molecular imprinting technology requires the formation of a polymer in the presence of a target template molecule. The monomers are mixed with template molecules, and this is followed by copolymerization. The electrochemical method can also be used to cause polymerization (electro-polymerization) [9,15]. After electro-polymerization, the template molecule is removed from the polymer matrix using solvent extraction. The removal of the template molecules results in the formation of three-dimensional receptor-type binding imprints with shapes and sizes complimentary to the template. The rebinding performance and thermodynamic properties of the imprints determine the overall utility of the sensor. The ortho-phenylenediamine used here is a monomer that is easily electro-polymerized and, hence, suitable as a molecular mold, providing easy-to-trap hydrophilic and hydrophobic target molecules [16]. Imprinted electrode-based o-phenylenediamine monomers have

been developed for the detection of oxytetracycline [17], sorbitol [18], and gibberellin A3 [19], among others. This is the first report of o-phenylenediamine monomers being used to make imprinted electrodes for oxycodone detection.

In this work, we discuss the preparation and electrochemical characterization of the oxycodone sensor. We focus on its analytical performance; in particular, its application to the quantitative analysis of oxycodone in wastewater samples. Oxycodone can be electrochemically oxidized on electrode surfaces and, hence, could interfere with the desired current signals. To overcome these potential current interferences, an electroactive molecule, $[Fe(CN)_6]^{3-/4-}$, with much lower redox potentials than oxycodone's oxidation potential was selected and used as a reversible redox probe. This redox probe was used as a reporter molecule. This probe competed with oxycodone for the imprinted recognition sites. When the sensor was immersed in oxycodone-containing solutions, the current obtained progressively decreased.

## 2. Materials and Methods

### 2.1. Chemicals

Oxycodone and morphine standards were purchased from VWR, while o-phenylenediamine was purchased from Sigma-Aldrich (St. Louis, MO, USA). Sodium acetate, acetic acid, potassium ferricyanide and potassium ferrocyanide were also purchased from VWR. For the present work, extra pure o-phenylenediamine was used without further purification. Deionized water was used throughout the study. All other aqueous solutions used here were prepared using deionized water.

Electrochemical Apparatus: Glassy carbon with a geometric area of 0.16 cm$^2$ was used for the working electrodes after functionalization using polymeric materials. The reference electrode was Ag/AgCl equipped with a glass tip. The glass tip was separated from the sample solution compartment by a salt-bridge made with potassium chloride and terminating in a medium porosity glass frit. Pt wire was used as a counter electrode. The reference and counter electrodes were purchased from Bioanalytical Systems Inc. (West Lafayette, IN, USA). The three electrochemical techniques, namely cyclic voltammetry (CV), square wave voltammetry (SWV) and differential pulse voltammetry (DPV), as well as electrochemical impedance spectroscopy (EIS) were carried out using a CHI instrument. This electrochemical workstation (CHI 660c, Austin, TX, USA) was computer controlled and had its ohmic drop (IR) 98% compensated.

### 2.2. Procedure Preparation of MIP Receptor

Glassy carbon electrodes were polished using standard cleaning procedures. This included polishing using a 1 μm diamond paste followed by ultrasonication in ethanol for one minute and in distilled water for another minute before drying in air. This final step included polishing with aluminum and rinsing with water. MIP electrodes were synthesized using electro-polymerization in a solution containing 2 μg/mL of oxycodone and 3.5 mM o-phenylenediamine. The mixture was stirred thoroughly before the electro-polymerization step. Electro-polymerization on a glassy carbon electrode was carried out by performing a cyclic voltammetry experiment at a 0.0 to 800 mV (vs. Ag/AgCl-saturated KCl) potential window and with Pt wire used as a counter electrode. The voltammogram sweeps were obtained at 100 mV/s. Important parameters influencing the polymerization include the number of cycles, o-phenylenediamine concentration, aggregate voltage range and template oxycodone concentration.

After electro-polymerization, the electrode with trapped oxycodone was rinsed with deionized water and then placed in a vial containing pure methanol solution for 30 min to extract the oxycodone trapped on the o-phenylenediamine polymer network. For comparison, another electrode was fabricated using the same protocol but omitting oxycodone during the polymerization step. The electrode thus produced was referred to as the non-imprinted polymer (NIP) electrode. Figure 2 shows a pictorial representation of the steps involved in this sensor construction.

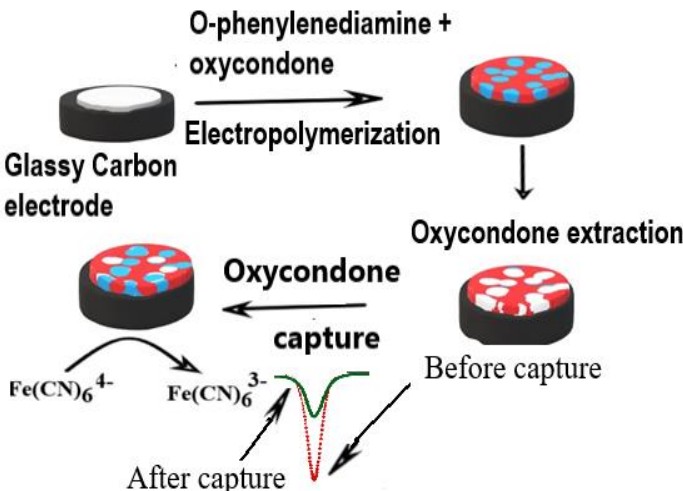

**Figure 2.** Pictorial representation of oxycodone detection using imprinted electrodes.

## 3. Results

### 3.1. Electrochemical Characterization of the Imprinted Sensor Electrodes

In the preliminary electrochemical study, the precise control of polymer thickness was established by carefully monitoring the electro-polymerization step. Investigation of the polymerization cycles was conducted to obtain the optimum electrochemical response. The thickness of the film depends on the number of cycles. The electro-chemical behavior of the synthesized polymer was obtained using cyclic voltammetry (CV) in 0.5 mM $[Fe(CN)_6]^{-3/-4}$ and 0.5 M acetate buffer solution at pH 5. Figure 3a shows a cyclic voltammogram of a layered growth of the o-phenylenediamine polymer formed after electro-polymerization. Sixteen cyclic voltammograms are shown. Figure 3b shows a comparison of a bare electrode with an electrode after polymer growth, both scanned in a 0.5 mM $[Fe(CN)_6]^{-3/-4}$ solution. The potential window was 0.0 V to 0.8 V, while the scan rate used was 0.1 V /S. It is clear from the red curve (Figure 3b) that the electron transfer process was seamless, while the polymer covering the conductive surface resulted in the complete blockage of electron transfer (blue curve). The formal potential $E^{o'}$ of the redox probe was unchanged at about 0.3 V. The electro-polymerization step was quite reproducible and hence provided a uniform film thickness. As shown in Figure 3a, the current decrease was negligible after 16 cycles. In the polymerization step, the concentration of o-phenylenediamine was maintained at 3.5 mM. A more rapid polymer formation was observed using a 7.5 mM o-phenylenediamine solution. However, the uniformity of the polymer was in question. Using 3.5 mM of the o-phenylenediamine monomer, 5 to 10 cycles were deemed as optimal conditions for electrochemical polymerization. The goal in this step was to electrodeposit a polymer that would enhance the formation of optimum prints without severely affecting the electron transfer.

### 3.2. Electrochemical Impedance Spectroscopy (EIS)

The effect of polymerization was also monitored by obtaining electrochemical impedance spectroscopy curves. Polymerization causes a change in the double layer capacitance, as well as the electron transfer resistance of the electrode. Changes introduced through electro-polymerization can be monitored by obtaining the electrochemical impedance spectroscopy (ESI) of the surfaces [20,21]. Although EIS is a technique developed some time ago [22,23], it has recently been gaining attention in electrochemical research, especially in the field of sensor development. EIS was used in this work to mainly monitor the growth of the polymer on the electrodes' surface. The EIS technique was carried out in a 0.5 mM $[Fe(CN)_6]^{-3}/[Fe(CN)_6]^{4-}$ in acetate buffer solution of pH 5.0 and performed using a frequency scan ranging from 100,000 to 1 Hz. The amplitude of the sine wave was 0.005 V, while the quiet time was set at 2 s.

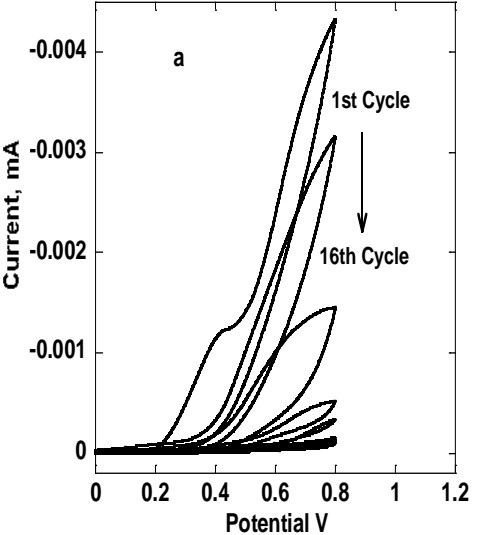
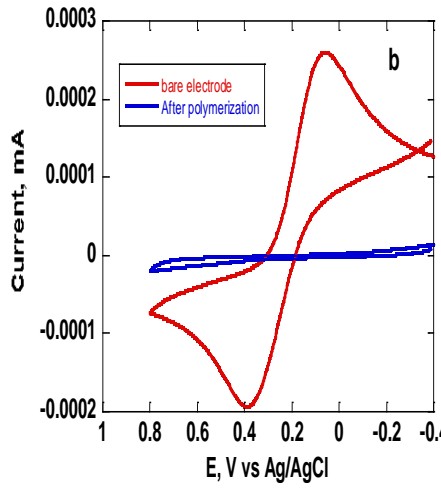

**Figure 3.** (**a**) Cyclic voltammograms showing the electro-polymerization of an o-phenylenediamine molecule on a glassy carbon electrode. (**b**) Cyclic voltammogram of a bare electrode compared with an electrode with that after polymerization, both scanned in a 0.5 mM Fe(CN)$_6$$^{-3/-4}$ solution.

Figure 4 shows the Nyquist plot (Z$'$ versus Z$''$) of different cyclic voltammogram scans representing different polymer thicknesses. The Randles equivalent circuit model (Figure 4 insert) used to fit the data is also shown, with Rs representing the electrolyte resistance, R$_{ct}$ representing the charge resistance, C$_{dl}$ representing the double layer resistance and Z$_w$ representing the Warburg impedance. The Cdl value changes when the thickness of the film is changed. The pattern of a simulated Nyquist plot following the Randles circuit is shown in Figure 4. The result is observed when all other parameters remain constant. It is obvious that increasing layers on the polymer film caused the complete disruption of the semicircle. The electrode ohmic resistance and the Cdl is the main cause of the impedance to the current flow. The semicircle diameter at higher frequencies in the Nyquist diagram reflects the interfacial electron transfer resistance (Rct) which controls the electron transfer kinetics of [Fe(CN)$_6$]$^{-3/-4}$ at the electrode surface. As the number of CV scans increased, the EIS resulted in an even larger diameter semicircle. The large semicircle indicates that the charge transfer kinetics are slower on the thicker electrode compared to that with a smaller number of polymer scans on the glassy carbon electrode.

Figure S1 shows the cyclic voltammogram (CV) of an imprinted electrode at different scan rates in a 0.5 mM [Fe(CN)$_6$]$^{-3/-4}$ in acetate buffer solution of pH 5.0. Peak current from cyclic voltammograms increased with scan rate as shown in Figure S2. The electrode was not exposed to oxycodone and therefore there was no blocking of the redox probe. The oxidation peak potential of [Fe(CN)$_6$]$^{-3/-4}$ was at 0.25 V, while the reduction peak was at 0.28 V. There was no significant shift in the peak potentials as the scan rate was increased, indicating that the Nernstian behavior was observed at the whole region of scan rate used. The anodic (I$_a$) and cathodic (I$_c$) peak current responses were equal, indicating reversibility. A plot of peak current (*i*) versus the square root of scan rates (*v*) indicates diffusion-controlled kinetics (Figure S3).

*3.3. Calibration Curves and Binding Affinity of Oxycodone MIP*

After the optimization of the number of voltammogram cycles and o-phenylenediamine concentrations, the sensor response towards different concentrations of oxycodone was investigated. To explore the optimization of the rebinding time, another round of experiments was carried out. The differential pulse voltammetry (DPV) current response of MIP electrodes incubated in a solution containing oxycodone at different times was obtained and the corresponding current–time curves were produced. In this experiment, a freshly made MIP electrode was produced and immersed in a 20 nM oxycodone solution to allow

rebinding. The electrode was removed from the oxycodone solution, rinsed with water then analyzed in a 0.5 mM $[Fe(CN)_6]^{-3/-4}$ probe solution. After again rinsing with water, the electrode was placed again in the same oxycodone solution for several minutes. The current signal was obtained every few minutes. Figure 5 shows the time-dependence plot of oxycodone rebinding on the MIP electrode. The magnitude of the observed current decreased with time. The current responses decreased and reached the minimum stable value after about 20 min. The molecule rebinding time is critical for a MIP sensor. Based on the DPV current responses, the optimum time for rebinding would be halfway between the plateau and the highest current which corresponds to around 60 μA of current or slightly less than 10 min of rebinding time. Using these parameters, calibration plots were produced to monitor the response of the oxycodone drug on both imprinted and non-imprinted electrodes using a 5 min incubation in 0 to 40 nM oxycodone concentrations at room temperature. The electrode was rinsed with methanol and water to remove loosely bound oxycodone molecules not properly absorbed at the surface. Cyclic voltammograms, as well as square wave voltammograms, of the electrode were obtained. The imprinted electrodes were placed in solutions containing different concentrations of oxycodone. SWVs were recorded in 0.5 mM $[Fe(CN)_6]^{-3/-4}$ for electrodes with different oxycodone concentrations. Figure 6a shows square wave voltammetry of imprinted electrodes in various concentrations of oxycodone. The peak current of the probe solution decreases with the increase in oxycodone concentration, indicating that the blocking effect of oxycodone was related the concentration. The rebinding of oxycodone to the imprint caused a reduced electron transfer.

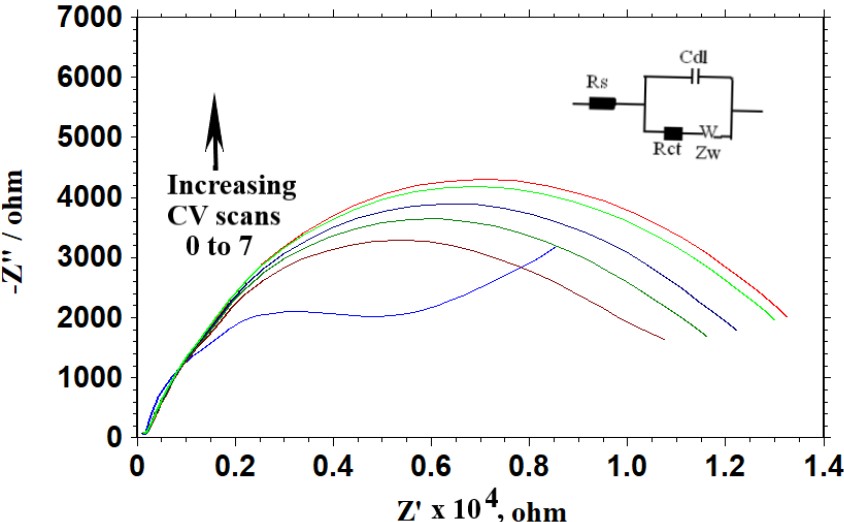

**Figure 4.** Electrochemical impedance spectroscopy of a glassy carbon electrode with continuous electro-polymerization. A spectra was taken after every cyclic voltammetry scan in $Fe(CN)_6^{-3/-4}$ solution. The blue spectrum represents an electrode without polymerization, while the red spectrum represents the 7th CV scan. The 1st to 6th CV scans are shown in different colors.

The SWV obtained with a zero oxycodone concentration showed the highest current. The MIP electrode placed in a 40 nM concentration of oxycodone presented the highest electrochemical redox process blocking effect as observed from the low peak currents of the redox probe. Figure 6b is a plot of peak current plotted against the concentration of oxycodone. A closer look at the plot reveals linearity in two regions. The first linear region is from 0.4 to 5 nM with the equation Y(Amps) = −0.000767 + 3.9835 × 10$^{-5}$ [oxycodone] with $R^2$ = 0.96, while the second region is from 10 to 40 nM with the equation (Y(amps) = −0.000585 + 2.0166 × 10$^{-6}$ [oxycodone], with $R^2$ = 0.998. The slope in the lower concentration region is much higher than in the higher concentration region, indicating a greater affinity of oxycodone molecules while the unoccupied sites are in excess. Above

10 nM oxycodone, the recognition sites of the MIP start to become saturated, resulting in smaller current decreases. The limit of detection (LOD) for oxycodone was 1.8 nM, calculated from (LOD = $3\sigma/m$), and the relative standard deviation (RSD) over three individual sensors was 0.239. On the other hand, the control experiment showed statistically insignificant changes in current over the whole range of oxycodone concentration. The non-imprinted polymer control shows that the nonspecific adsorption of oxycodone does not play a significant role in the observed sensitivity judging from the linear background current.

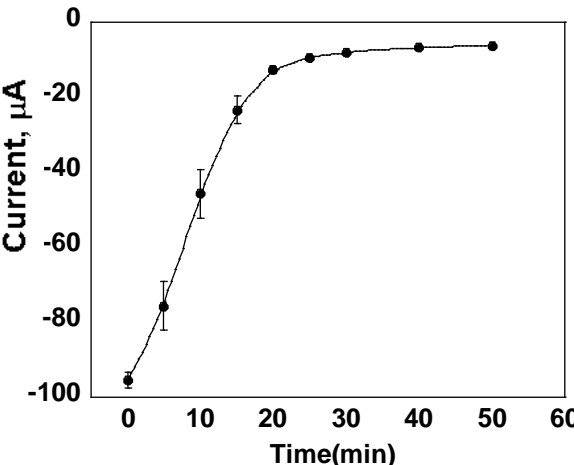

**Figure 5.** Time-dependence plot of oxycodone rebinding on an imprinted electrode monitored as a function of time.

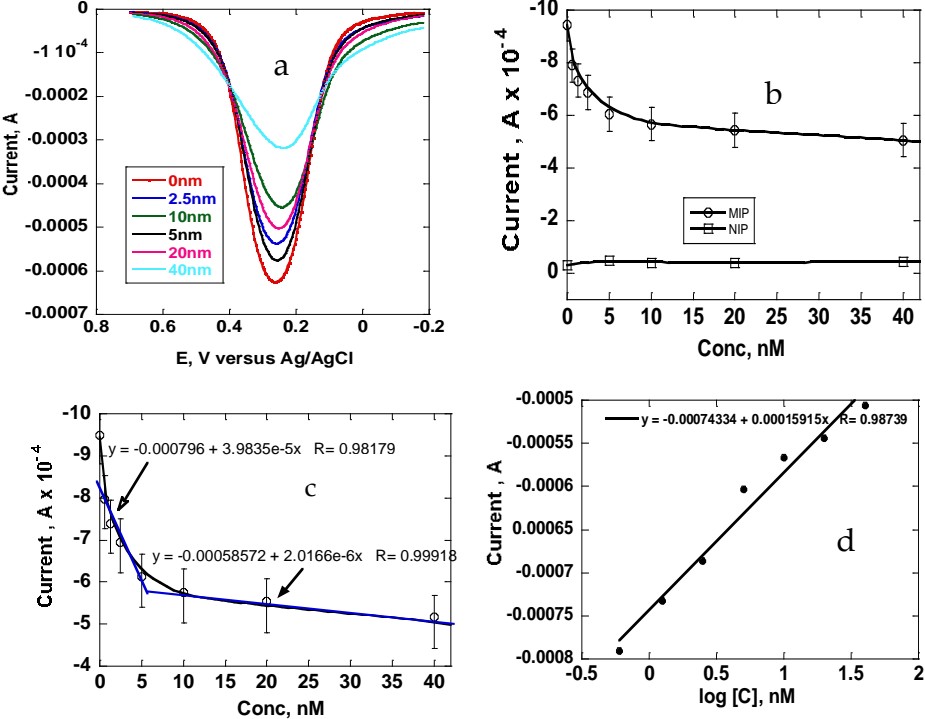

**Figure 6.** (**a**) Square wave voltammetry of o-phenylenediamine polymer imprinted electrodes analyzed in different concentrations of oxycodone solution. (**b**) Current as a function of concentration of both imprinted and non-imprinted electrodes. (**c**) Current as a function of concentration, showing the two linear regions. (**d**) The calibration curve of current against the logarithmic values of the oxycodone concentrations.

Using the calibration plots between the current and the logarithmic value of oxycodone concentration from 0 to 40 nM also demonstrated a good linear relationship. The regression equation was I (A) = −0.0007433 + 0.000159log[oxycodone] (nM) with a correlation coefficient ($R^2$) of 0.98. This sensor for oxycodone detection was compared with those from other literature. This sensor shows a respectable sensitivity compared to other sensors (Table 1). Obviously, those sensors that use or involve chromatographic methods have a higher sensitivity but as noted earlier, they are more expensive and complicated.

**Table 1.** Comparisons of different methods for the detection of oxycodone.

| Electrode-Based | | | |
| --- | --- | --- | --- |
| **Modified Electrodes** | **Linear Range** | **Detection Limit** | **Reference** |
| $CoFe_2O_4$/CPE | 0.06–38 uM | 0.05 uM | [24] |
| Nafion/SWCNT | 0.5–10 μM | 85 nM | [25] |
| o-Phenylenediamine polymer imprint | 0.4–5 nM | 1.8 nM | This work |
| Other methods | | | |
| HPLC-UV | 1–2000 ng/mL | 2.67 ng/mL | [26] |
| nano-Si-based SALDI-MS. | 0–100 ng/mL | 1.56 ng/mL | [27] |
| ELISA combine with GC-MS | 0 to 1000 ng/g | 50 ng/g | [28] |
| Silver NPs on zinc oxide, SERS | 900 ug/mL to 90 ng/mL | 90 ng/ mL | [29] |

The responses of both the MIP and NIP electrodes sensors are useful in estimating the affinity of the imprinted sites and oxycodone molecules. The square wave data shown in Figure 6a were used to study the binding properties of the MIP.

A plot was made with the *x*-axis being the change in the probe current response ($i_0 - i$), while the *Y* axis was the normal oxycodone concentration. The difference between the current recorded in the presence of oxycodone and the current recorded in the absence of the oxycodone was denoted as i and $i_0$, respectively. The binding isotherm of oxycodone onto the MIP-electrode binding sites was fitted using a Langmuir model. This Langmuir binding isotherm model assumed that there was an equilibrium between oxycodone and the MIP electrode binding sites. Figure 7 shows the fitted theoretical curve of the data. The number of binding sites available to bind oxycodone molecules determines the current observed and hence the differences between the peak currents recorded in the absence and the presence of oxycodone. We used the following mathematical expression obtained from a modified Langmuir equation [30].

$$i_0 - i = \frac{[oxy]k_A BS}{1 + [oxy]K_A}$$

Peak currents in the absence and presence of oxycodone are represented by $i_0$ and i, respectively. The total number of binding sites is represented by BS. The association constant $K_A$ represents the equilibrium between oxycodone and the binding sites in the MIP. $K_A = 1.12 \times 10^6$ for the isotherm, which indicated the high affinity of oxycodone molecules to associate with the binding sites.

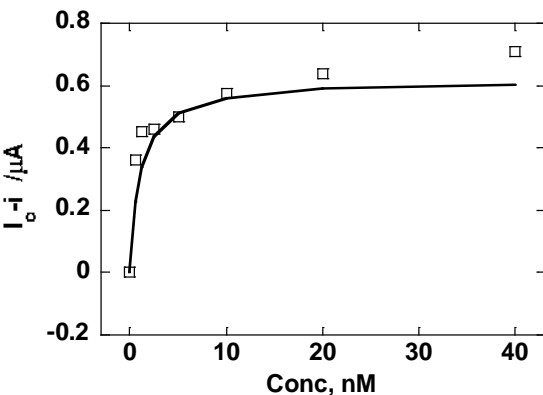

**Figure 7.** Experimental data from the 0.5 mM $[Fe(CN)_6]^{-3/-4}$ in an acetate buffer system fitted to a Langmuir model for the adsorption of oxycodone into the MIP recognition sites.

### 3.4. Interference Study

Common interferants in the analysis of oxycodone in wastewater or in blood include other drug metabolites such as noroxycodone and oxymorphone. Other opioids could also interfere with the measurement The selectivity of this sensor was evaluated by analyzing morphine. Morphine and oxycodone are closely related in terms of their structure and chemical properties. As shown in Figure 8a, the current changes obtained with oxycodone are much higher compared to those from morphine, indicating that the sensor is selective towards oxycodone.

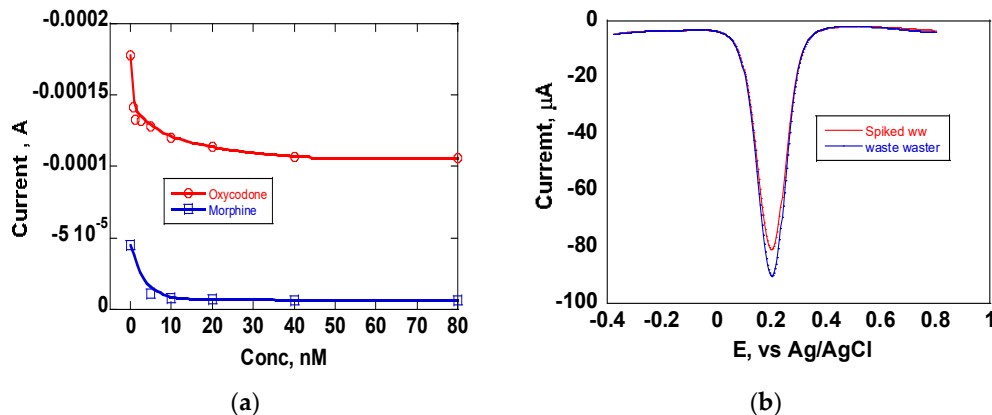

(a)                                                                                         (b)

**Figure 8.** (**a**) Current as a function of the concentration of oxycodone using imprinted electrodes. Comparison of oxycodone and morphine re-binding (**b**) DPV of oxycodone using imprinted electrodes in a wastewater sample compared with the electrode in wastewater spiked with 20 nM oxycodone.

Finally, these new imprinted sensor electrodes were used to test wastewater samples collected from a nearby city. To test these samples, the wastewater was filtered and kept refrigerated until needed. One of the filtered samples was split into two. One of the samples was spiked with oxycodone to provide a solution that is 6 nm. To the other solution, deionized water was added to keep the volume the same. The imprinted electrodes were immersed in these solutions for 5 min and later rinsed with water as usual before measurement. Figure 8b shows the DPV of the imprinted electrodes comparing the two waste-water samples. The results show a value of about 90 μA for the un-spiked wastewater. The current was reduced to 80 μA after spiking with oxycodone. This indicated that the wastewater spiked with oxycodone encountered more blocking than the un-spiked one, as would be expected. Wastewater contains many molecules and materials that can interfere with the measurement, but it appears that an appreciable current is obtained in this sensor.

## 4. Discussion

The application of MIPs to opioid detection has not been widely demonstrated judging from the few studies in the literature on the topic. Oxycodone is a molecule that can be oxidized on an electrode surface. The oxidation potential of oxycodone is based on the material used in the electrode. In SWCNT, the oxidation potential is about 0.9 V versus a Ag/AgCl electrode [25]. In this work, the oxidation potential using glassy carbon would be much higher than 0.9 V and the concentration needed would be much more than the 5 uM previously used for SWNT material. Using a potassium ferrocyanide probe with an oxidation potential of less than 0.4 V on glassy carbon helped to avoid the direct oxidation of oxycodone. The polymerization step was a critical step, as each scan contributed a layer of polymer that significantly reduced electron transfer. It was important to have a three-dimensional polymer to support imprint formation at the same time as allowing electron transfer. Increasing the concentration of o-phenylenediamine or increasing the number of scans served the same purpose. In our case, 5 to 10 CV scans of 3.5 mM of o-phenylenediamine gave optimal results. The formation of imprints was a success, as the electrodes for NIP showed a complete block of electron transfer. The MIP electrode placed in a 20 nm oxycodone solution at different times showed current blocking being a function of time up to 20 min when no changes in current occurred. There was no evidence that oxycodone molecules detached from the imprints in an equilibrium manner. This would explain the short time scale of oxycodone rebinding on the imprinted electrode. The MIP electrodes were found to have a higher sensitivity towards oxycodone at lower concentrations. This meant that the rebinding at the lower concentration was more efficient, indicating a greater affinity toward oxycodone molecules to the binding sites. Compared with other electrode-based oxycodone sensors in the literature, this sensor has a lower detection limit and hence is useful in the determination of wastewater samples. Wastewater samples have a low concentration of opioids due to dilution from point sources. We have also included other sensors that use or involve chromatographic methods that obviously have a higher sensitivity and a wide linear range but as noted earlier, these are more expensive and complicated. The performance of this sensor is related to the strong association of the oxycodone and the binding site. The MIP-modified electrode's binding isotherm when fitted using a Langmuir model showed a high association constant, corroborating the high affinity of oxycodone molecules to associate with binding sites.

These results demonstrate that these oxycodone-imprinted electrodes possessed an excellent ability to bind oxycodone in imprinted cavities. This sensor is much simpler to fabricate than others found in the literature. The proposed strategy has universal significance in creation of opioid-molecule-imprinted polymers for binding specific opioids.

## 5. Conclusions

In this contribution, the preparation and characterization of MIP electrodes for electrochemical detection oxycodone was demonstrated. The o-phenylenediamine monomer mixed with oxycodone molecules followed by electro-polymerization and extraction caused the formation of receptor-type binding imprints with shape and size complimentary to oxycodone. The rebinding performance was used to analyze for oxycodone. The sensor with a linear range from 0.4 to 5 nM with a limit of detection of 1.8 nm has a short response time with complete saturation at about 20 min. Therefore, these results suggest that MIP electrode sensors have utility in the rapid screening of oxycodone in water samples. The presence of interfering molecules such as morphine has a statistically insignificant effect. This technique can be applied to fabricate other MIP sensors for other opioids.

**Supplementary Materials:** The following supporting information can be downloaded at: https://www.mdpi.com/article/10.3390/electrochem4040028/s1, Figure S1: Plot of scan rate dependence of an imprinted electrode in $Fe(CN)_6^{+3/4}$ solution. Cyclic voltammograms with increased scan rate; Figure S2: A plot of current versus scan rate obtained from S1; Figure S3: A plot of current versus square root scan rate obtained from S1.

**Author Contributions:** Conceptualization, methodology, validation and original draft preparation was performed by A.M. Formal analysis, investigation and data curation was performed by P.C. Review and editing, visualization, supervision and project administration were performed by A.M. All authors have read and agreed to the published version of the manuscript.

**Funding:** This research received no external funding.

**Institutional Review Board Statement:** Not applicable.

**Informed Consent Statement:** Not applicable.

**Data Availability Statement:** Data are available upon request from the corresponding author.

**Acknowledgments:** The authors would like to acknowledge Rowan University for the Research Fellowship provided, as well the departmental facilities that enabled this work.

**Conflicts of Interest:** The authors declare no conflict of interest.

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
