# Peer review of "Molecularly Imprinted Electrochemical Sensor Based on Poly (O-Phenylenediamine) for Sensitive Detection of Oxycodone in Water"

_2673-3293, doi:10.3390/electrochem4040028_

Round 1

Reviewer 1 Report

1.    Author should describe more about the health concern problems of oxycodone in the introduction section.

2.    Authors requested to add pictorial representation of oxycodone detection with proper representation.

3.    In section 3.1, author mentioned “the thickness of the film depended on the number of cycles” if possible, provide some characterization study to find out the thickness of the electrode.

4. Author should give the chemical reaction involved in the electrochemical detection of oxycodone.

5.    Authors have not provided the pH study, so it is need to provide the study for different range of pH then the suitable pH should be taken as consideration for further experiments.

6.    In the figure 6a, author should describe the cause of the non-linear current increment with addition of different concentrations of oxycodone.

7.    In table 1, Authors requested to add two separate columns for modified electrodes and techniques.

8.    Authors mentioned in the introduction part “oxycodone is occurred in the blood and urine of patients” author should use the human blood and human urine for practicability application of the electrode instead of waste water.

9.    In interference studies, authors requested to add different opioids which interfering the detection of oxycodone.

10. Author should care the proper representation of the x axis and y axis for different figures in main manuscript and supplementary.

English language should be improved..

Author Response

Thanks for taking the time to review this manuscript. Your comments are very helpful not only in this manuscript but also in some future manuscripts. 

Reviewer 2 Report

Following are my suggestions.

1. Some sort of characterization ( spectroscopical or microscopic) is necessary to prove the presence of electropolymerized oxycodone at the surface of the electrode.

2. Please repeat the experiment for Figure 6. The signal in the linear range ( 0.4 to 5 nm) is too close. The error bars are not very clear ( Figure 6C). The current does not seem to change from 5 nM to 20 nM and the error bar is too high! Where is the error bar in the log plot?

Author Response

Thanks for Taking time to review this manuscript

Reviewer 3 Report

Manuscript ID: electrochem-2518117

I have read the manuscript entitled “Molecularly imprinted electrochemical sensor based on Poly (o-phenylenediamine) for sensitive detection of oxycodone in water”. It aims at the fabrication of the MIP-modified electrode for the detection of oxycodone with nanomolar. The manuscript cannot be accepted in its current form. The author should focus on the scientific significance of your work. However, the authors still have a chance to revise their manuscript.

Detail points that should be considered.

(1)   The author needs to provide a structural and morphological characterization of MIP formation.

(2)   “Glassy carbon (0.16 cm2) was used as working electrodes”, Did the author mention the geometrical surface (0.16 cm2) area of the working electrode? Give a clear explanation.

(3)   The molecular formula of the redox probe ([Fe(CN)6]3-/4−) is not mentioned uniformly throughout the manuscript so the author should check and correct the mistakes. Also, focus on the superscript, subscript, and scientific units.

(4)   Line 158, “in in”, delete the repeated words. “A 7.5 mM concentration resulted in a more rapid polymer formation”. Revise the sentence.

(5)   According to EIS results, the author needs to explain why the semicircle portion was increased but the linear portions disappeared due to the increased CV scans (0 to 7). Detail explanation is needed. Mention the Rct values of all scans.

(6)   The author should revise the supplementary information file figures and captions. Include the experimental conditions in figure captions. 

(7)   Why did the author choose the 0.5 mM [Fe(CN)6]3-/4- in acetate buffer solution of pH 5.0 as a supporting electrolyte in this work? The oxycodone doesn’t have any response in the acetate buffer? Why don't you study pH optimization? 

(8)   In Line 220, the author wrote “The oxidation peak potential was at 0.25 V while the reduction peak was at 0.28 V”. Is the redox peak potential corresponding to oxycodone or 0.5 mM [Fe(CN)6]3-/4-?

(9)   What was the procedure used to detect the oxycodone? DPV or SWV? The author should revise the manuscript.  

(10)  Table 1 needs to revise. Column 1 is not the technique. Remove the full stop in the title of the manuscript.

(11)  How about the stability of MIP modified electrode? In Fig. S3, the author should mention the X-axis unit correctly.

(12)  The author should pay more attention and provide scientific information to explain the results. English also needs to improve. 

The authors need to improve the quality of the English Language.

Author Response

Thanks for taking your time to review this manuscript. Some of the questions you raise will help as we think about the next phase of the work.

Round 2

Reviewer 2 Report

The error bar is still too high in the calibration curve.

It is okay

Author Response

Thanks for taking time to review this manuscript. Your comments will definitely help us as we tailor next experiments. 

Comment: The error bar is still too high in the calibration curve.

Response: You are correct, the error bars are high. We were concerned about this. It appeared that some small variability occurred each time new electrodes were made contributing to the significant standard deviation. Although we were unable to bring  down the standard deviation, it appears this is not uncommon with some imprinted electrodes, see the literature below.

ACS Appl. Mater. Interfaces 2011, 3, 2, 191–203

Reviewer 3 Report

The author should pay more attention and provide scientific information to explain the results.

English needs to improve. 

Author Response

Thank you for taking time to review this manuscript. We appreciate your comments to enable us to improve. 

Comment: The author should pay more attention and provide scientific information to explain the results.

Response: We have made changes to the manuscript on some sections. We have also revised it to improve the English.